# Mental health outcomes and intimate partner violence among nepalese women: A propensity score matched study

Geoffrey Barini[1]*, Linnet Ongeri[2], Polycarp Mogeni[2]

1 Department of Pure and Applied Mathematics, Jomo Kenyatta University of Agriculture and Technology, Nairobi, Kenya, 2 Kenya Medical Research Institute (KEMRI), Nairobi, Kenya

* barini@jkuat.ac.ke

## Abstract

Although intimate partner violence (IPV) is an established risk factor for mental health disorders, the impact of the various types of IPVs is not well understood. We examined the risk of exposure to IPV on generalized anxiety and depression among Nepalese women. Using data from the 2022 Nepal Demographic and Health Survey, we performed a 1:1 propensity score matching restricted to women aged 15–49 with current or former intimate partner. The modified Poisson regression was used to assess the risk of anxiety or depression symptoms in women exposed to IPV, and E-values assessed sensitivity to unmeasured confounding. Of 4,245 women enrolled in the study, 350 (8.2%) had generalized anxiety, 259 (6.1%) had depressive disorders, and 1264 (29.8%) reported having experienced at least one form of IPV. Women exposed to any form of IPV were associated with a higher risk of anxiety (Adjusted risk ratio [aRR]=1.51; [95%CI: 1.17-1.96]; p=0.002) and depression symptoms (aRR=2.56; [95%CI: 1.92-3.40]; p<0.001). The highest risk of depressive symptoms was observed among women who reported emotional or sexual IPV occurring together with male controlling behavior (aRR between 4.98 and 7.00; p<0.001). However, the associations with generalized anxiety symptoms were modest across all forms of IPV co-occurring with male controlling behavior. That is, aRR=1.84; ([95%CI: 0.87-3.90]; p=0.111) for Physical IPV, aRR=1.93; ([95%CI: 1.06-3.51]; p=0.032) for Emotional IPV, and aRR=1.21; ([95%CI: 0.61-2.39, p=0.588]) for Sexual IPV. IPV is associated with an increased risk of generalized anxiety and depressive symptoms among Nepalese women. Sexual or physical violence co-occurring with male controlling behaviors poses higher mental health challenges, suggesting that male controlling behaviors may signify severe or prolonged abuse. Our findings support comprehensive interventions addressing mental health needs and the root causes of IPV.

**Data availability statement:** The DHS data used in this study is available at https://dhsprogram.com/data/available-datasets.cfm and can be freely accessed by registered users.

**Funding:** The authors received no specific funding for this work.

**Competing interests:** The authors have declared that no competing interests exist.

## Introduction

Intimate partner violence (IPV) is a serious global public health issue and a fundamental violation of human rights [1]. IPV refers to behaviors by an intimate partner that inflict physical, sexual, or psychological harm, including physical aggression, sexual coercion, psychological abuse, and controlling actions [1]. Women are disproportionately affected, particularly in low- and middle-income countries (LMICs) such as Nepal, where approximately 27% of ever-partnered women reported experiencing physical, sexual, or emotional violence from a current or former intimate partner [2].

IPV is associated with significant short- and long-term health consequences, including physical injuries, mental health disorders, unwanted pregnancies, and sexually transmitted infections, and in some cases death [3–7]. While much of the evidence on IPV's mental health-related effects comes from high-income countries, studies from LMICs have demonstrated an association between IPV and mental health disorders, including depression, anxiety, and suicide [1,5,8,9]. These consequences may be even more severe in Nepal due to the stigma surrounding mental health and harmful societal norms that normalize violence against women [10–13].

Evidence suggests a link between IPV and the development of mental disorders such as anxiety and depression among Nepalese women [14,15]. However, this fails to differentiate the impact of various IPV subtypes. For instance, one study estimated that 20% of women with IPV experience depressive symptoms [16]. Another study involving an analysis of nationally representative data demonstrated that approximately half of women exposed to domestic violence had symptoms of depression and anxiety [14]. Moreover, findings from the 2022 demographic and health survey pointed out that only 7% of women exhibiting anxiety and depressive symptoms reported seeking medical attention [2]. Although IPV has been consistently identified as an important predictor of a wide range of mental health disorders, evidence indicates that a poor social support system, being young, single, unemployed, or having witnessed inter-parental IPV increases one's likelihood of developing anxiety and depressive symptoms [15,17–20].

Although the relationship between mental health and IPV has been shown in some settings, the magnitude of the associations has been variable. Current evidence on the effects of IPV on mental health is generally weak and often limited by methodological constraints. Furthermore, very few studies have considered male controlling behaviours both independently and their co-occurrence with other forms of IPVs in regression models. In addition, the potential correlation between various forms of IPVs is often overlooked.

Using propensity score matching, we aimed to investigate whether exposure to the individual forms of IPVs and their co-occurrence with male controlling behaviours are associated with the risk of symptoms of generalized anxiety and depression among Nepalese women aged 15–49 years.

## Methods

### Study design and participants

Here we present a secondary data analysis using the 2022 Nepal demographic and health survey (NDHS) [21]. The Demographic and Health Surveys (DHS) are

nationally representative surveys conducted by government agencies in LMIC with support from the U.S Agency for International Development (USAID). The DHS uses standardized questionnaires and sampling methods to collect comprehensive information on country-specific demographics and health data, including domestic violence, mental health, and women empowerment among others [21].

The 2022 NDHS followed a two-stage stratified cluster sampling design to select survey participants. First, all seven provinces of Nepal were segmented into urban and rural categories, creating 14 sampling strata. Within each stratum, probability-proportional-to-size (PPS) sampling was applied. In the first stage, 476 primary sampling units (PSUs) were selected with PPS, including 248 PSUs from urban areas and 228 from rural areas. Within each PSU, geographic clusters, defined as enumeration areas (EAs) containing approximately 200 households were identified. A household listing exercise was conducted in all selected PSUs to establish the sampling frame for the second stage. Subsequently, a simple random sample of 30 households was selected from each EA, resulting in 14,280 households for the survey.

Female interviewers invited all eligible women aged 15–49 from selected households to participate in face-to-face interviews, ensuring a nationally representative sample of households and women of reproductive age. Of the 15, 238 eligible women 14, 845 completed the interviews, resulting in a 97.4% response rate. Further detailed information on the sampling procedure and selection to participate in the domestic violence and mental health modules has been published elsewhere [2]. Our analysis included women who met the following criteria: (1) women who participated in 2022 NDHS, (2) aged 15–49 years, (3) who were selected for the domestic violence module, and (4) who have or ever had an intimate partner.

## Exposure variables

In the main analysis, exposures were: (1) any form of intimate partner violence, including physical, emotional, and sexual violence. (2) Physical violence, (3) emotional violence, (4) sexual violence, (5) co-occurrence of physical and sexual violence, (6) co-occurrence of physical and emotional violence, (7) co-occurrence of emotional and sexual violence, and (8) co-occurrence of all forms of IPV. In the ancillary analyses, we sought to assess the magnitude of the co-occurrence of controlling behaviors with physical, emotional, or sexual violence. We defined exposure to each form of IPV as a binary variable (1 = exposure, 0 = no exposure).

This study considered physical, emotional, and sexual violence and controlling behavior perpetrated by the current or former intimate partner within 12 months preceding the 2022 NDHS. IPV-related information was collected using the modified version of the conflict tactics scale [22]. Physical IPV-related acts included being *"violently shaken, pushed or had an object thrown at her"*, *"slapped"*, *"punched with a fist or hit by something that could hurt"*, *"kicked, dragged or beaten"*, *"chocked or burnt on purpose"*, *"threatened with a knife, gun or any other weapon"* and *"had her arm twisted or hair pulled"*; emotional IPV-related acts included being *"humiliated"*, *"threatened with harm"*, and *"insulted or made to feel bad"*; sexual IPV-related acts included being *"physically forced to have unwanted sex"*, *"physically forced to perform unwanted sexual acts"* and *"forced with threats or any other way to perform unwanted sexual acts"*; male controlling behaviors included *"does not permit her to meet her female friends, tries to limit her contact with her family and insists on knowing where she is at all times"*.

## Outcome variables

The study considered three binary outcomes: a) symptoms of any mental disorder defined as a composite indicator consisting of symptoms of generalized anxiety or depression, b) symptoms of generalized anxiety, and c) depressive symptoms. Symptoms of generalized anxiety and depression were assessed using the Generalized Anxiety Disorder scale (GAD-7) and the Patient Health Questionnaire (PHQ-9), respectively. GAD-7 and PHQ-9 are widely validated and recognized as effective screening instruments for identifying individuals at increased risk of mental disorders in both primary care settings and the general population [23,24].

For anxiety, respondents were asked how often they had experienced any of the following symptoms two weeks preceding the survey: *'feeling nervous, anxious or on edge', 'not being able to stop or control worrying', 'worrying too much about different things', 'trouble relaxing', 'being so restless that it is hard to sit still', 'becoming easily annoyed or irritable', 'feeling afraid as if something awful might happen'*. The frequency of these symptoms was scored on a four-point Likert scale, ranging from 0 (never) to 3 (always). The total score was used to assess the severity of anxiety symptoms.

Similarly, to assess depressive symptoms, respondents reported the frequency with which they had experienced the following symptoms two weeks preceding the survey: *'little interest or pleasure in doing things', 'feeling down, depressed or hopeless', 'trouble falling asleep, staying asleep, or sleeping too much', 'feeling tired or having little energy', 'poor appetite or overeating', 'feeling bad about yourself - or that you're a failure or have let yourself or your family down', 'trouble concentrating on things, such as reading the newspaper or watching television', 'moving or speaking so slowly that other people could have noticed or, the opposite - being so fidgety or restless that you have been moving around a lot more than usual, 'thoughts that you would be better off dead or of hurting yourself in some way'*. Responses were also scored on a four-point Likert scale, from 0 (never) to 3 (always), with the total score ranging from 0 to 27, used to assess the severity of depressive symptoms. Medical diagnosis of depression or anxiety by a qualified healthcare practitioner were captured using two questions: "*Have you ever been told by a doctor (been diagnosed) that you have depression*?" and "*Have you ever been told by a doctor (been diagnosed) that you have anxiety*?" with binary responses "yes/no".

Having symptoms of generalized anxiety was defined as GAD-7 ≥ 8 whilst depression was defined as PHQ-9 ≥ 10 [23,24]. The validated PHQ-9 cut-off values for Nepalese adolescent and adult populations are presented in S1 Table [2]. Respondents with a prior diagnosis of anxiety or depression from a doctor, and who were taking medication for either condition, were classified as positive for symptoms of anxiety or depression, regardless of their GAD-7 or PHQ-9 scores.

## Statistical analysis

In this propensity score-matched population-based analysis, we sought to estimate the risk of symptoms of generalized anxiety and depressive disorders among Nepalese women who are exposed to intimate partner violence. We performed a 1:1 propensity score matching (PSM) using the 'genetic algorithm' and a caliper of width 0.1, with exact matching on prognostic variables to create comparable treatment groups based on selected potential confounders [25,26]. The confounders were selected based on prior studies and the success of balancing distributions between exposed and unexposed groups [27–31]. The socio-demographic variables used for matching were age, level of education, region, self-reported health status, income status, marital status, substance use, household food insecurity, severe disability, pregnancy loss, and alcohol drinking pattern of the intimate partner. Exact matching was performed on self-reported health status, marital status, and household-level food insecurity [32–34]. A detailed description of these variables is presented in (S2 Table).

To ensure robust comparisons, we controlled for exposure to all forms of intimate partner violence (IPV) within our analysis by only comparing each IPV type (excluding co-occurrence) with the control group. Our comparison (control) group was defined to include only women who reported no experience of any form of IPV. This approach differs from earlier studies that compared women exposed to one form of IPV to groups potentially exposed to other forms of IPV [15], likely confounding the independent relationship between individual IPV and mental health outcomes.

Propensity scores were estimated using the logistic regression model, and the quality of covariate balance between exposure groups before and after PSM was assessed by calculating the absolute standardized mean differences (ASMD). The absence of residual imbalance (ASMD>=0.1) in all covariates was indicative of a successful PSM [35]. To address potential misspecification in the models due to residual imbalance in covariates, we used the "doubly robust" method in which all covariates used in propensity score estimation and the exposure variable were included in the outcome regression analyses [36]. Given the low proportion of missing data (less than 1.4%), we did not perform multiple imputation.

## Sensitivity analyses

We conducted several sensitivity analyses. First, we used GAD-7 and PHQ-9 cutoff values specific to Nepal [2] to determine risk estimates for anxiety and depressive disorders associated with any form of IPV; second, we repeated this analysis without propensity score matching; third, we excluded women who had given birth within the past 12 months prior to the survey. Finally, we assessed the robustness of our conclusion to unmeasured confounders using the E-values [37].

Adjusted risk ratios (aRR) and the 95% confidence interval (95%CI) were estimated using the modified Poisson regression. Statistical analyses were two-sided, with p-values < 0.05 deemed statistically significant, and were performed using R software version 4.3.3 (https://www.r-project.org/).

## Ethical statement

The 2022 NDHS protocol was developed and reviewed by the Nepal Health Research Council and the ICF Institutional Review Board [2]. The data is anonymized and publicly available for researchers who request access by completing an online request form available on the DHS website [21]. Therefore, no ethical approvals were required for the preparation of the manuscript [2]. Strengthening the Reporting of Observation Studies in Epidemiology (STROBE) statement checklists were followed to guide transparent reporting of the study [38].

# Results

## Demographic characteristics of the study participants

A total of 5,178 women were enrolled in the domestic violence module, with 4,245 having a current or former intimate partner. Most women (90.8%) were married or cohabiting with an intimate partner (S3 Table). Among the respondents, 36.7% had attained secondary or higher education, 47.7% were employed all year round, and 23.5% reported no employment. The proportion of respondents who reported severe disability was 5.8%, while 10.3% self-reported poor health, and more than a third (38.8%) reported experiencing household food insecurity.

## Prevalence of IPV and symptoms of mental disorders

The overall prevalence of experiencing any form of IPV among Nepalese women was 29.8% (1264 women). In this subsample, male controlling behavior was the most common form of IPV (43.4%) among women experiencing only one type of IPV (excluding co-occurring IPV types) followed by physical IPV (12.3%) and emotional IPV (7.8%) (S4 Table). When examining women experiencing male controlling behavior and one other type of IPV, the prevalence of physical IPV was 17.1% (216 women), emotional IPV was 18.9% (239 women), and sexual IPV was 8.5% (107 women).

Among those reporting any IPV, 145 (11.5%) had symptoms of generalized anxiety, and 138 (10.9%) had depressive symptoms. Of the respondents exposed to sexual IPV and controlling behaviors, 33 (30.8%) reported experiencing symptoms of depression. Additionally, the proportion of women who experienced symptoms of depression was higher than those who experienced symptoms of anxiety in the subsample containing each IPV (physical, emotional, or sexual IPVs) co-occurring with male controlling behaviours (S5 Table). Estimates based on <25 unweighted samples were excluded due to concerns about statistical reliability, following DHS reporting guidelines [39].

## Propensity score matching

After estimating propensity scores and performing 1:1 genetic matching [25], 1263 of 1264 women exposed to any type of IPV were successfully matched, resulting in a total sample size of 2526. For the individual types of IPVs (excluding co-occurrence), the total sample size after PSM was 1066 for male controlling behaviours, 304 for physical IPV, and 194 for emotional IPV. We did not match respondents exposed to sexual IPV due to insufficient sample size (Fig 1). PSM matching results for physical, emotional, and sexual IPVs co-occurring with controlling behaviours are shown in Fig 1. The

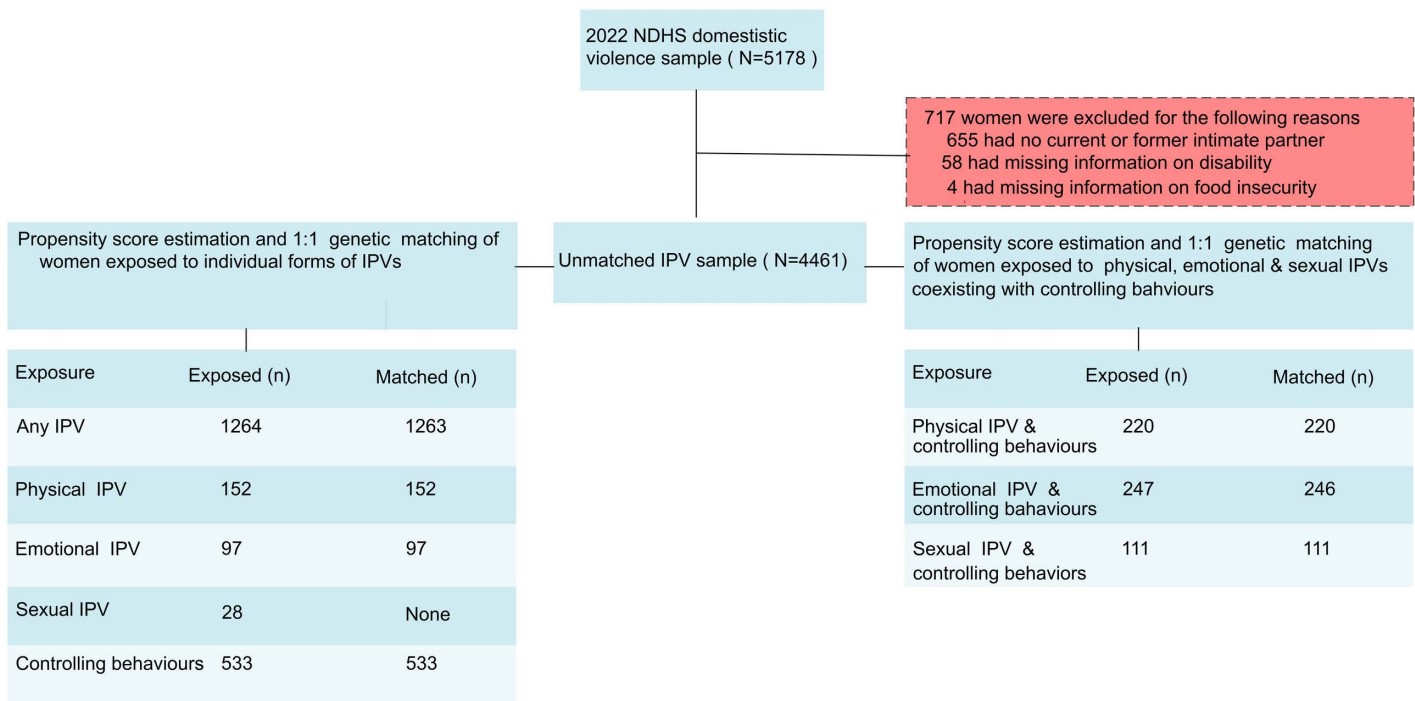

**Fig 1. A flowchart illustrating the number of individuals included in the PSM analyses.** IPV-exposed women were matched 1:1 to controls using genetic propensity score matching. Due to insufficient sample size, women exposed to sexual violence alone were not included in the matching process. The left side of the flowchart displays IPV types excluding co-occurrence with other forms of IPV. The right side presents participants who reported experiencing male controlling behaviors in addition to one other type of IPV. Comparison groups were defined as women who reported no experience of any type of IPV.

standardized mean differences for the selected demographic characteristics were less than 0.1 (Figs 2–3) and (S6–S9 Tables), indicating good balance for each potential confounder.

### Risk of generalized anxiety and depressive disorder

In the propensity score-matched analysis, exposure to any type of IPV increased a woman's risk of generalized anxiety or depressive disorder by 88% (aRR = 1.88; [95%CI: 1.57-2.25]; p < 0.001) (Fig 4). In an analysis of each outcome separately, exposure to any type of IPV increased a woman's risk of generalized anxiety and depressive disorders by 51% (aRR = 1.51; [95%CI: 1.17-1.96]; p = 0.002) and >2-fold (aRR = 2.56; [95%CI: 1.92-3.40]; p < 0.001) respectively (Fig 4). In the analyses of individual IPVs, without co-occurrence, only emotional violence was associated with an elevated risk of depressive symptoms (aRR = 2.42; [95%CI: 1.09-5.36]; p = 0.032) (Fig 4). Strikingly, physical, sexual or emotional IPV co-occurring with male controlling behaviours was strongly associated with poor mental health outcomes (Fig 4). That is, when each IPV (physical, emotional, or sexual) occurred together with male controlling behaviours, the risk of symptoms of depression was > 4-fold (aRR between 4.74 and 7.00; p < 0.001) (Fig 4). However, their associations with the risk of generalized anxiety symptoms were modest, with adjusted risk ratio between 1.21 and 1.93 (Fig 4).

### Sensitivity analyses

Our findings were consistent across all sensitivity analyses (S10 and S11 Tables), including results from the unmatched analyses (S12 Table). Furthermore, the PSM analysis results were consistent when Nepal-specific cutoff values for generalized anxiety and depressive disorders were used.

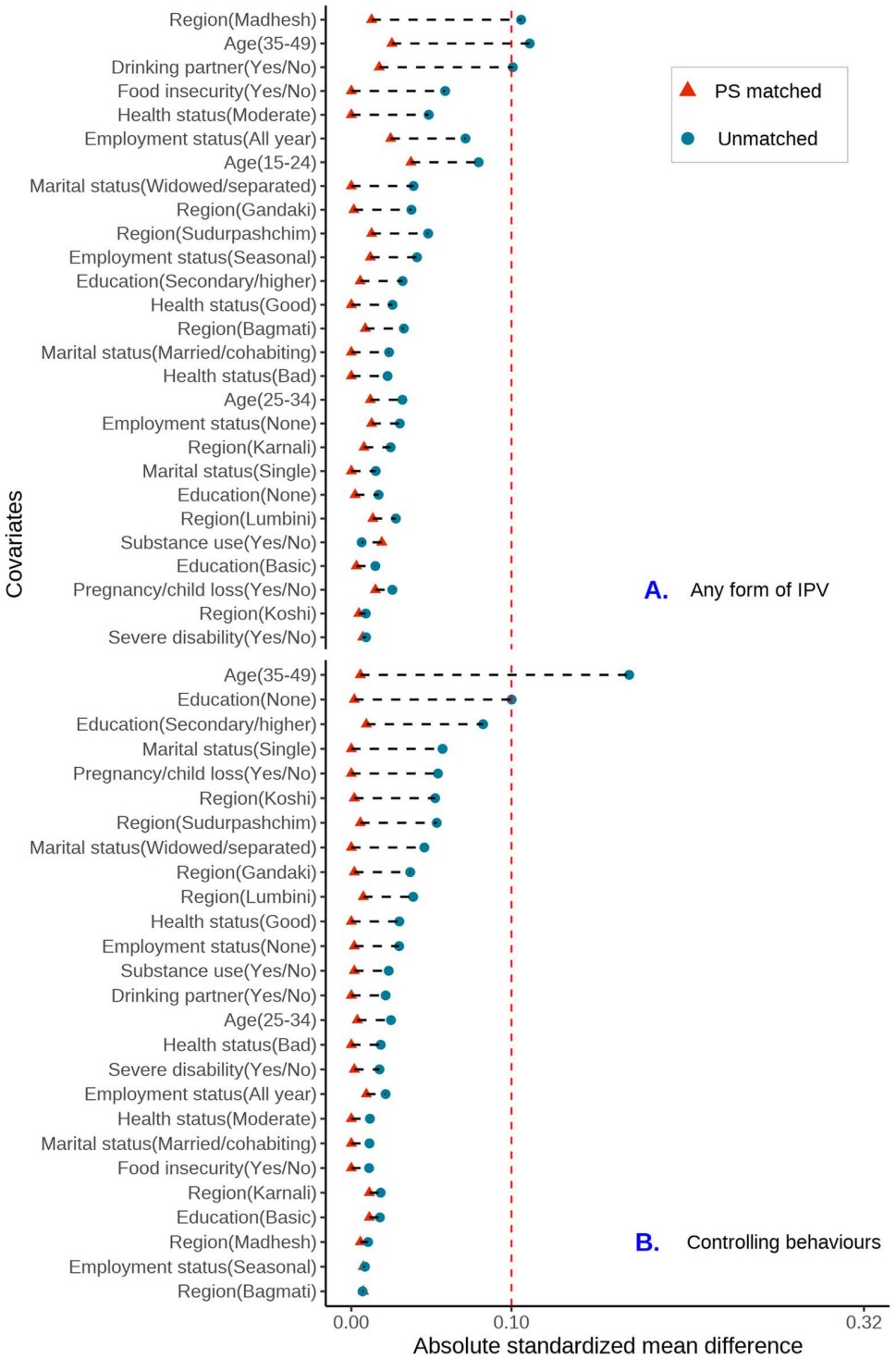

**Fig 2. Covariate balance love plots of the absolute standardized mean differences for any form of IPV and male controlling behaviours before and after propensity score matching.**

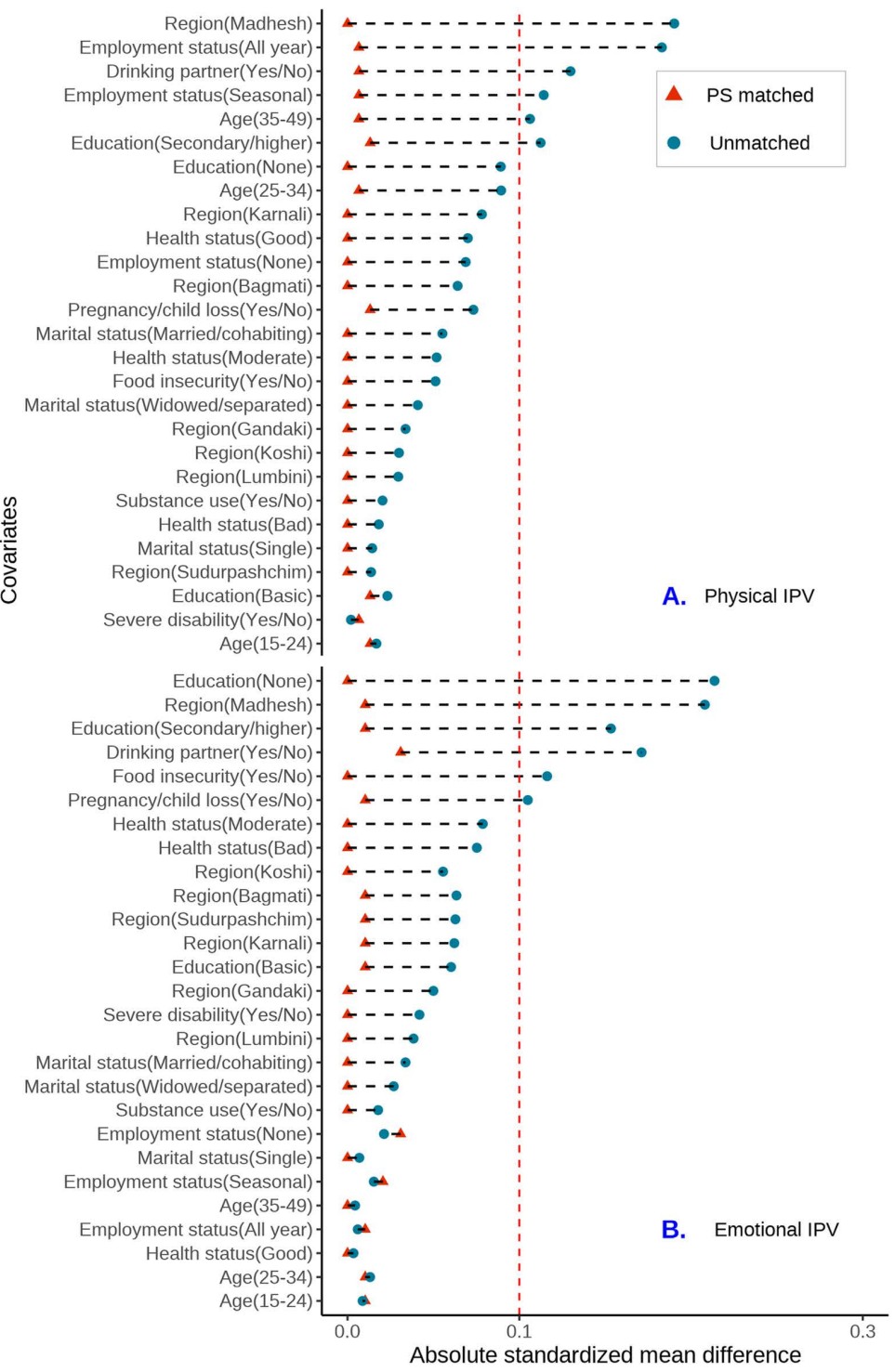

**Fig 3. Covariate balance love plots of the absolute standardized mean differences for Physical and emotional IPVs before and after propensity score matching.**

| Exposure | Outcome | aRR ( 95%CI ) | P Value | |
|---|---|---|---|---|
| Any IPV | Symptoms of anxiety or depression | 1.88 ( 1.57-2.25 ) | <0.001 | |
| | Symptoms of anxiety | 1.51 ( 1.17-1.96 ) | 0.002 | |
| | Symptoms of depression | 2.56 ( 1.92-3.40 ) | <0.001 | |
| Physical IPV | Symptoms of anxiety or depression | 0.95 ( 0.56-1.60 ) | 0.847 | |
| | Symptoms of anxiety | 0.96 ( 0.43-2.12 ) | 0.915 | |
| | Symptoms of depression | 1.09 ( 0.53-2.24 ) | 0.810 | |
| Emotional IPV | Symptoms of anxiety or depression | 1.81 ( 0.93-3.51 ) | 0.084 | |
| | Symptoms of anxiety | 2.00 ( 0.88-4.58 ) | 0.102 | |
| | Symptoms of depression | 2.42 ( 1.09-5.36 ) | 0.032 | |
| Controlling behaviours | Symptoms of anxiety or depression | 1.20 ( 0.85-1.68 ) | 0.297 | |
| | Symptoms of anxiety | 1.04 ( 0.65-1.66 ) | 0.873 | |
| | Symptoms of depression | 1.67 ( 0.91-3.08 ) | 0.099 | |
| Physical IPV and controlling behaviours | Symptoms of anxiety or depression | 3.07 ( 2.04-4.62 ) | <0.001 | |
| | Symptoms of anxiety | 1.84 ( 0.87-3.90 ) | 0.111 | |
| | Symptoms of depression | 4.74 ( 2.54-8.83 ) | <0.001 | |
| Emotional IPV and controlling behaviours | Symptoms of anxiety or depression | 3.07 ( 2.10-4.49 ) | <0.001 | |
| | Symptoms of anxiety | 1.93 ( 1.06-3.51 ) | 0.032 | |
| | Symptoms of depression | 4.98 ( 2.75-9.01 ) | <0.001 | |
| Sexual IPV and controlling behaviours | Symptoms of anxiety or depression | 3.13 ( 2.31-4.24 ) | <0.001 | |
| | Symptoms of anxiety | 1.21 (0.61-2.39 ) | 0.588 | |
| | Symptoms of depression | 7.00 ( 4.52-10.85 ) | <0.001 | |

Adjusted Risk Ratio (aRR)

**Fig 4. Estimates of adjusted risk ratios of the associations between individual types of IPVs and their co-occurrences with male controlling behaviors, and mental health outcomes among Nepalese women using PSM modified Poisson regression.**

To assess the robustness of our findings against unmeasured confounding, E-values for each exposure and corresponding mental health outcomes for which statistically significant associations were observed were calculated (S13 Table). Among women experiencing any form of IPV, the E-value was 3.12, indicating that unmeasured confounding would need to be strongly associated with both IPV and generalized anxiety or depression to account for the observed magnitude of risk. Similarly, the E-values of any IPV were substantial (>2.39 for anxiety, >4.56 for depression) when anxiety and depression were analyzed separately. These results were consistent in the subgroup analyses involving individual types of IPV and their co-occurrence with intimate partner controlling behavior, E-values >3 (S13 Table).

## Discussion

Our results demonstrate a large negative association of exposure to IPV on mental health outcomes among Nepalese women of reproductive age. First, our results indicate that women exposed to any type of IPV were associated with an 88% increase in symptoms of generalized anxiety or depression. Second, an analysis of each outcome separately, albeit with substantial comorbidity, demonstrates that exposure to IPV was associated with a >2-fold increase in symptoms of depression and a 51% increase in symptoms of generalized anxiety. Third, subgroup analyses examining each type of IPV individually revealed that only emotional IPV was significantly associated with depression. However, further analyses demonstrated that individual IPV types co-occurring with intimate partner's controlling behaviors were more strongly associated with adverse mental health outcomes compared to instances where each IPV type occurred in isolation. Taking these findings together with previously reported findings [5,40–43], we conclude that women's exposure to IPV strongly predicts mental health outcomes and that male-controlling behaviors are likely a strong indicator of the severity of exposure. Strategies to reduce mental health challenges should focus not only on treatment programs but also on IPV mitigation through education on care seeking and harmful societal norms that continue to perpetuate IPV [1].

The observed variations in the associations between different types of IPV (excluding co-occurrences) and depressive symptoms may be attributable to varying levels of IPV severity. These findings align with previous research demonstrating a positive correlation between IPV severity and poorer mental health outcomes. For instance, a study conducted in Bangladesh found that the risk of major depressive episodes increased significantly with increasing IPV severity [44]. Corroborating this, studies conducted in high-prevalence settings, particularly in low- and middle-income countries, have consistently reported a weaker or non-existent association between less severe forms of IPV and depressive symptoms [45,46]. Nepal faces significant societal challenges related to deeply ingrained patriarchal norms [12,13]. These norms contribute to the normalization of violence against women and perpetuate systemic gender-based discrimination [12,13,47]. This societal context may partially explain the observed lack of associations between some types of individual IPV and anxiety symptoms. That is, in settings where discrimination against women is normalized, women may be more likely to internalize or accept these experiences, potentially leading to a reduced likelihood of experiencing symptoms of anxiety as a direct psychological response [44]. Alternatively, the inconsistency in the strength of the findings from each type of IPV may be attributed to the decreased statistical power resulting from the subgroup analyses (excluding co-occurrence). Therefore, further research is warranted to explore these contextual factors and their implications on mental health outcomes.

In contrast, women exposed to a combination of male controlling behaviors and at least one other type of IPV (physical, emotional, or sexual) were associated with a substantially higher risk of depressive symptoms. The concurrent presence of male controlling behaviors and other forms of intimate partner violence (IPV) may indicate greater IPV severity or length of victim exposure. In addition, women in relationships characterized by both controlling behaviors and other forms of violence may be associated with significant social isolation and a lack of adequate support. Social isolation can severely limit coping mechanisms, thereby increasing vulnerability to depressive symptoms [1,30,48,49].

Our study has important strengths. First, few studies provide a comprehensive assessment of the potential associations of IPV with mental health outcomes among Nepalese women using the PSM and nationally representative surveys. Second, most previous studies assessing the effect of IPV on mental health outcomes have compared women exposed to one form of IPV with women exposed to other forms, obscuring the independent effects of each form of IPV and their co-occurrence with male-controlling behaviors [15]. When the independent effects of each type of IPV on mental health outcomes are desired, this approach may yield biased results. In our study, the independent effects of various types of IPV were assessed against a comparison group defined to include only women who reported no experience of any form of IPV.

This study is limited by its reliance on self-reported cross-sectional data, which may be subject to biases such as social desirability bias, recall bias, and underreporting due to the sensitive nature of IPV [50]. Underreporting of IPV experiences may result from stigma surrounding IPV as well as normalized or socially acceptable abuse within intimate relationships [10,12,13]. In addition, despite the robust methodology deployed (PSM), the possibility of unmeasured confounding cannot be entirely ruled out. However, the E-value analysis suggests that any such confounder would need a markedly strong association with both IPV exposures and mental health outcomes, with a relative risk exceeding 2, to significantly impact the observed findings. Furthermore, a comprehensive literature review did not establish any variables with such a strong and plausible association. This study was conducted in Nepal; a population that experiences significant societal challenges related to deeply ingrained patriarchal norms and discriminatory practices against women and, as such, may be difficult to generalize more broadly except to those countries with cultural similarities. Our analyses relied on cross-sectional survey data, which inherently limits our ability to determine the temporal sequence between IPV and mental health outcomes, making it challenging to definitively establish the causal direction. However, evidence from several longitudinal studies suggests a consistent and causal link between exposure to IPV and mental health disorders [51–54]. Finally, sub-setting the data to create clean cases and control groups can inherently reduce statistical power, potentially leading to an increased risk of type II errors. Therefore, the lack of statistically significant associations between physical

IPV and male controlling behaviors, excluding co-occurrence, with mental health outcomes may be attributable, in part, to reduced statistical power. However, a study investigating the association between IPV subtypes and HIV infection found that neither physical IPV nor male controlling behaviors, when considered in isolation, were significant predictors of HIV infection [3]. This finding suggests that our results are unlikely to be due to insufficient statistical power. In conclusion, our findings highlight the need for a shift towards comprehensive, multifaceted interventions that address both IPV prevention and the treatment of mental health outcomes. This approach should be feasible in resource-limited settings, extend beyond traditional symptom-focused treatment, and provide a holistic and effective response to mental health challenges in similar contexts [1]. Mental health and IPV programs should consider identifying and mitigating barriers to women's access to available support services, and ensure the availability of effective referral systems to community-based care to these interventions [1,55–58]. These programs should incorporate strategies to promote mental health literacy, reduce stigma, and foster positive health-seeking behaviors for common mental health disorders [1].

## Supporting information

**S1 Table. Nepal anxiety and depression cut-off scores for adolescent and adult population.**
(DOCX)

**S2 Table. Potential confounders used in the study.**
(DOCX)

**S3 Table. Social demographic characteristics of the study population.**
(DOCX)

**S4 Table. Survey weighted prevalence estimates of the individual types of IPVs and their co-occurrence with male controlling behavior in the unmatched sample.**
(DOCX)

**S5 Table. Survey weighted prevalence estimates of symptoms of generalized anxiety and depression by type of IPV.**
(DOCX)

**S6 Table. Covariate balance for physical violence co-occurring with male controlling behavior.**
(DOCX)

**S7 Table. Covariate balance for emotional violence co-occurring with male controlling behavior.**
(DOCX)

**S8 Table. Covariate balance table for sexual violence co-occurring with male controlling behavior.**
(DOCX)

**S9 Table. Covariate balance for any violence excluding women who gave birth 12 months prior to the survey.**
(DOCX)

**S10 Table. PSM univariable and multivariale modified Poisson regression outcome analysis results for women exposed to any IPV using Nepal GAD-7 and PHQ-9 Cut-off values.**
(DOCX)

**S11 Table. PSM univariable and multivariale modified Poisson regression analysis results excluding women who gave birth within 12 months prior to the survey.**
(DOCX)

**S12 Table. Univariable and multivariable modified Poisson regression analysis results for women exposed to any IPV for the unmatched sample.**
(DOCX)

**S13 Table. E-values for point estimates and lower 95% confidence limits for outcomes for which there was observed associations with IPV.**
(DOCX)

**S1 Checklist. PRISMA Checklist.** PRISMA 2009 checklist.
(DOTX)

## Author contributions

**Conceptualization:** Geoffrey Barini, Polycarp Mogeni.

**Data curation:** Geoffrey Barini.

**Formal analysis:** Geoffrey Barini, Polycarp Mogeni.

**Investigation:** Geoffrey Barini, Linnet Ongeri, Polycarp Mogeni.

**Methodology:** Geoffrey Barini, Polycarp Mogeni.

**Project administration:** Polycarp Mogeni.

**Resources:** Geoffrey Barini, Polycarp Mogeni.

**Software:** Geoffrey Barini.

**Supervision:** Geoffrey Barini, Polycarp Mogeni.

**Validation:** Geoffrey Barini, Polycarp Mogeni.

**Visualization:** Geoffrey Barini, Polycarp Mogeni.

**Writing – original draft:** Geoffrey Barini.

**Writing – review & editing:** Geoffrey Barini, Linnet Ongeri, Polycarp Mogeni.

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
