## [Decision Letter · Decision Letter 0]

PMEN-D-25-00054

Mental Health Outcomes and Intimate Partner Violence among Nepalese Women: A Propensity Score Matched Study

PLOS Mental Health

Dear Dr. BARINI,

Thank you for submitting your manuscript to PLOS Mental Health. After careful consideration, we feel that it has merit but does not fully meet PLOS Mental Health’s publication criteria as it currently stands. Therefore, we invite you to submit a revised version of the manuscript that addresses the points raised during the review process.

We look forward to receiving your revised manuscript.

Kind regards,

Gellan Karamallah Ramadan Ahmed

Academic Editor

PLOS Mental Health

Journal Requirements:

1. Please upload a copy of Figure 1-4 which you refer to in your text on page 22-25. Or, if the figure is no longer to be included as part of the submission please remove all reference to it within the text.

2. We have noticed that you have a list of Supporting Information legends in your manuscript. However, there are no corresponding files uploaded to the submission. Please upload them as separate files with the item type 'Supporting Information'. 

Additional Editor Comments (if provided):

Reviewers' comments:

Reviewer's Responses to Questions

**Comments to the Author**

1. Does this manuscript meet PLOS Mental Health’s publication criteria?

Reviewer #1: Yes

Reviewer #2: Yes

2. Has the statistical analysis been performed appropriately and rigorously?

Reviewer #1: Yes

Reviewer #2: Yes

3. Have the authors made all data underlying the findings in their manuscript fully available (please refer to the Data Availability Statement at the start of the manuscript PDF file)?

Reviewer #1: Yes

Reviewer #2: No

4. Is the manuscript presented in an intelligible fashion and written in standard English?

Reviewer #1: Yes

Reviewer #2: Yes

Reviewer #1: This is an interesting and important paper examining mental health and IPV in Nepal.

The manuscript is well written and clear, therefore I only have minor comments.

The authors are affiliated with institutions in Kenya, was there anyone from Nepal involved as this is usually good practice to understand the context.

Analysis:

Were any weights used to adjust for non-response etc? Please outline what they accounted for.

What was done to deal with missing data? What was the response rate for the survey.

Minor points:

- Typo in Figure 1 red box - inoformation

-Line 273 reference error

-Line 241 - refers to 1 person exposed to sexual violence. I don't know if the DHS has any rules about reporting small numbers but the authors may want to reconsider reporting this due to potential sensitivities.

Reviewer #2: The study uses a cross-sectional design, which limits its ability to determine cause and effect. The propensity score matching only controls for variables that were measured, leaving room for unaccounted confounders. Self-reported data on IPV and mental health can lead to underreporting and recall bias. The small sample size for certain IPV types reduces the power to detect significant associations. Overall, these methodological issues weaken the reliability of the study’s conclusions. Addressing these issues can significantly improve the study.

**Do you want your identity to be public for this peer review?** For information about this choice, including consent withdrawal, please see our Privacy Policy

Reviewer #1: No

Reviewer #2: No

---

## [Decision Letter · Decision Letter 1]

PMEN-D-25-00054R1

Mental Health Outcomes and Intimate Partner Violence among Nepalese Women: A Propensity Score Matched Study

PLOS Mental Health

Dear Dr. BARINI,

Thank you for submitting your manuscript to PLOS Mental Health. After careful consideration, we feel that it has merit but does not fully meet PLOS Mental Health’s publication criteria as it currently stands. Therefore, we invite you to submit a revised version of the manuscript that addresses the points raised during the review process.

We look forward to receiving your revised manuscript.

Kind regards,

Gellan Karamallah Ramadan Ahmed

Academic Editor

PLOS Mental Health

Journal Requirements:

Additional Editor Comments (if provided):

Reviewers' comments:

Reviewer's Responses to Questions

**Comments to the Author**

Reviewer #1: All comments have been addressed

Reviewer #3: All comments have been addressed

Reviewer #4: (No Response)

publication criteria?

Reviewer #1: Yes

Reviewer #3: Yes

Reviewer #4: Yes

3. Has the statistical analysis been performed appropriately and rigorously?

Reviewer #1: I don't know

Reviewer #3: I don't know

Reviewer #4: Yes

4. Have the authors made all data underlying the findings in their manuscript fully available (please refer to the Data Availability Statement at the start of the manuscript PDF file)?

Reviewer #1: Yes

Reviewer #3: Yes

Reviewer #4: Yes

5. Is the manuscript presented in an intelligible fashion and written in standard English?

Reviewer #1: Yes

Reviewer #3: Yes

Reviewer #4: Yes

Reviewer #1: Thank you for addressing my comments.

I have a minor suggestion for the abstract - to include full numerical results for the figures given in the final sentence of the findings section as already provided for the other numerical results.

On Line 264 the link has an error.

Reviewer #3: All the comments have been addressed by the authors. Yet, I had a very minor suggestion in Line 375, Heading is "Role the funding source" which is grammatically incorrect. You may consider it to change into, "Funding source."

Line 376, Since there is no funding source, only mention about, "The DHS had no role in..." there is no need to mention about the role of funders.

If this minor thing is addressed, I believe the study is good to go for the publication in Plos Mental Health. Good luck

Reviewer #4: The methodology section describes using propensity score matching (PSM) to control for measured confounders and E-values to address unmeasured confounding. While PSM is mentioned in the Methods (abstract), the E-value analysis is omitted. I recommend adding a brief reference to E-values in the abstract (e.g., ‘Sensitivity analyses with E-values assessed robustness to unmeasured confounding’) to align with the full methodology.

**Do you want your identity to be public for this peer review?** For information about this choice, including consent withdrawal, please see our Privacy Policy

Reviewer #1: No

Reviewer #3: No

Reviewer #4: No

---

## [Decision Letter · Decision Letter 2]

Mental Health Outcomes and Intimate Partner Violence among Nepalese Women: A Propensity Score Matched Study

PMEN-D-25-00054R2

Dear DR. BARINI,

We are pleased to inform you that your manuscript 'Mental Health Outcomes and Intimate Partner Violence among Nepalese Women: A Propensity Score Matched Study' has been provisionally accepted for publication in PLOS Mental Health.

Best regards,

Gellan Karamallah Ramadan Ahmed

Academic Editor

PLOS Mental Health

Reviewer Comments (if any, and for reference):

Reviewer's Responses to Questions

**Comments to the Author**

Reviewer #1: All comments have been addressed

Reviewer #3: All comments have been addressed

Reviewer #4: All comments have been addressed

publication criteria?

Reviewer #1: Yes

Reviewer #3: Yes

Reviewer #4: Yes

3. Has the statistical analysis been performed appropriately and rigorously?

Reviewer #1: Yes

Reviewer #3: Yes

Reviewer #4: Yes

4. Have the authors made all data underlying the findings in their manuscript fully available (please refer to the Data Availability Statement at the start of the manuscript PDF file)?

Reviewer #1: Yes

Reviewer #3: Yes

Reviewer #4: Yes

5. Is the manuscript presented in an intelligible fashion and written in standard English?

Reviewer #1: Yes

Reviewer #3: Yes

Reviewer #4: Yes

Reviewer #1: No further comments

Reviewer #3: No comments remaining. I have reviewed the previous submissions as well as the Revised V.2 of the manuscript. The comments are well addressed and ready to go for publication.

Good Luck

Reviewer #4: (No Response)

**Do you want your identity to be public for this peer review?** For information about this choice, including consent withdrawal, please see our Privacy Policy

Reviewer #1: No

Reviewer #3: **Yes: ** Rehnuma Abdullah

Reviewer #4: No
